# The COVID-19 Pandemic and Its Impacts on Tourism Business in a Developing City: Insight from Vietnam

**Da Van Huynh** [1,*] , **Thuy Thi Kim Truong** [1], **Long Hai Duong** [1,2,*] , **Nhan Trong Nguyen** [1],
**Giang Vu Huong Dao** [1] and **Canh Ngoc Dao** [1]

1    School of Social Sciences and Humanities, Can Tho University, Can Tho 900000, Vietnam;
     kimthuy@ctu.edu.vn (T.T.K.T.); trongnhan@ctu.edu.vn (N.T.N.); daovhgiang@gmail.com (G.V.H.D.);
     dncanh@ctu.edu.vn (C.N.D.)
2    Department of Global Hospitality and Tourism, Kyung Hee University, Seoul 02447, Korea
*    Correspondence: hvda@ctu.edu.vn (D.V.H.); long.7032450@gmail.com (L.H.D.)

**Abstract:** The COVID-19 pandemic has generally destroyed the global tourism industry and threatened the recovery of destinations in developing countries facing more challenges from increasingly serious waves of the pandemic. Although many studies have attempted to measure the general impacts of COVID-19, very little research has been conducted to assess its overall impact on specific tourism destinations throughout many waves of the pandemic. This research aims to explore how a tourism economy in a developing country context has been damaged after many waves of COVID-19. A typical emerging city in Vietnam experiencing three waves of the COVID-19 pandemic was selected as a case study. The study recruited 40 representatives of tourism-related organizations for in-depth interviews, while 280 questionnaires were distributed to participants from different tourism organizations. The findings indicate that the majority of tourism businesses in the examined case study seriously suffered from the pandemic, and very few tourism-related enterprises were able to recover after the first wave of infection. Unfortunately, the tourism business sectors were found to be on the brink of bankruptcy or facing permanent shutdown after the third wave. All tourism enterprises generally appeared to experience a sharp drop in the number of customers, tourism revenue, service facilities and exploitation, as well as employee downsizing, but the degree of downturn differed among the examined enterprises. Among the tourism enterprises, travel agencies and the accommodation sector were found to suffer the greatest economic losses compared to other stakeholders. In general, the COVID-19 pandemic's impact on the tourism business in Vietnam is a big concern, which may require a timely economic policy response and financial scheme to better support local enterprises in coping with the challenges during post-pandemic recovery.

**Keywords:** COVID-19; pandemic; tourism; impact; Vietnam

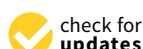



## 1. Introduction

Tourism generates huge revenue for destinations and contributes to overall economic development for a plethora of countries, especially tourism-dependent nations. In other words, there is a strong link between tourism development and economic growth, which also stimulates the development of other related businesses in a country (Haryanto 2020). However, tourism is a vulnerable industry which may crash due to potential risks such as global pandemics (Shakya 2009). The COVID-19 pandemic has inflicted serious and widespread impacts on a wide range of economic sectors, particularly the tourism industry. This is likely true for countries where the economy mainly depends on tourism, where businesses face much bigger challenges for economy resilience in both the short and long term. An abundance of consequences, including regional to national lockdowns, as well as the suspension or cancellation of tourism- and hospitality-related services, have increasingly occurred on the global scale. A typical comparison is that the COVID-19 pandemic has been estimated to be nine times more devastating than the impact of the

September 11 crisis; the economic revenue loss was enormous, at around USD 124 billion in 2020 alone (Haryanto 2020; Sarkodie and Owusu 2021). Unfortunately, the profound and long-lasting impacts of the COVID-19 pandemic may damage global economies, especially businesses in developing countries where the economy depends on tourism, and where the level of tourism resilience after pandemics is limited (Ismael et al. 2021).

In Vietnam, the COVID-19 pandemic and its impact on the tourism industry are major concerns due to multiple lockdowns imposed since 2020. The first case of COVID-19 infection in Vietnam was detected on 23 January 2020. Tourism activities in Vietnam in general and Can Tho city in particular fell into a state of crisis and almost total paralysis until March 2021. Most of the services and activities associated with tourism and hospitality were stopped or interrupted, which led to devastating consequences, including a loss of revenue for the tourism industry and a loss of jobs for workers. From June 2020, the domestic tourism industry started to recover; however, with the impact of the second and third waves of COVID-19, the situation once again became a struggle, with a difficult economic recovery from the pandemic anticipated. According to the International Monetary Fund (IMF), some economic sectors in Vietnam were projected to be severely affected, especially tourism, transportation and accommodation services. According to the annual estimated statistics, Can Tho's tourism industry generates a value of nearly VND 4500 billion and creates jobs for nearly 40,000 people. Following multiple lockdowns, the recovery of the city's tourism industry remains in a difficult and ineffective state due to the lack of a scientific basis to accurately assess the current situation and propose proper recovery solutions.

To some extent, the timely policies and supportive measures have contributed to the alleviation of the impact of the pandemic and the prevention of mass bankruptcy for local businesses. Therefore, it is essential and urgent for the tourism industry in Can Tho city to assess the current status of COVID-19's impact and find proper solutions for post-crisis recovery. As such, this research aimed to explore how the COVID-19 pandemic has influenced the tourism industry in Can Tho city, and identify the key tourism sectors and the degree to which local tourism businesses have suffered from waves of the pandemic. Therefore, the purpose of this study was to examine two research questions: (1) How hard has the COVID-19 pandemic hit the tourism industry in a developing city in Vietnam? and (2) How vulnerable are the local tourism enterprises in response to the pandemic's impacts?

## 2. Literature Review

This section reviews key studies regarding the economic impact of the COVID-19 pandemic on the tourism industry on a global scale, particularly reflecting on current research in Vietnam to provide a better understanding of how tourism businesses in a developing country context have suffered economically from the lingering pandemic crisis period.

The hospitality sector across different regions of the world has also suffered enormously due to the pandemic. Nguyen (2020) claimed that large hospitality enterprises are likely to be more resilient to the effects of the pandemic. This means that small businesses are at disadvantage, facing a risk of shutdown or even bankruptcy. In addition, the aviation sector has been vulnerable to the pandemic, with continuous lockdowns and flight restrictions leading to airline bankruptcies on a global scale (Dube et al. 2021). This has been evidenced by large airlines around the world facing bankruptcy and uncertainties as a result of the pandemic (Gole et al. 2021). Furthermore, the accommodation sector has been affected by the COVID-19 pandemic, whereby "new normal" measures have resulted in hotels adapting to maintain their business (Krouk and Almeida 2021). Lastly, tourist attractions have faced challenges with the potential scenario of shutdowns following repetitive waves of the COVID-19 pandemic (Prihadi et al. 2021).

The World Health Organization (WHO) officially declared COVID-19 a pandemic on the 11th of March 2020. The pandemic was projected to cause a wide range of socio-cultural,

political and economic impacts (Sigala 2020). Obviously, the global tourism industry is highly vulnerable to such crises, and the impacts on the global tourism destinations will be profound and long-lasting (Rassy and Smith 2013). Evidently, the significant 78% decrease in the global tourist arrivals rate and loss of around 120 million jobs were accounted to lead to a loss of USD 1.2 trillion in export revenues (Sigala 2020; UNWTO 2020). This indicates that global economic losses due to the pandemic are enormous, but particularly in destinations whose economy system mainly depends on tourism businesses.

According to Ulak (2020), global mobility has spread the pandemic to all countries and the tourism intention among the global tourists has dropped significantly for safety- and health-related reasons. Hoque et al. (2020) found that tourists were generally cautious when making trips during outbreaks of the pandemic. More seriously, the authors reported that psychological impacts of the pandemic on tourist's travel intentions and experiences during the crisis were a major concern. Similarly, other studies also highlighted the negative impacts of the COVID-19 pandemic on different sectors of the tourism and hospitality industries across different countries.

The tourism industry in Asia has also suffered enormously during the initial hits of the COVID-19 pandemic. More specifically, unemployment, bankruptcy, revenue loss and budget deficits are common serious consequences brought about by the COVID-19 pandemic (Kasare 2020). In the Philippines, the impact of COVID-19 on the national tourism business was estimated to represent a loss of more than USD seven billion as the pandemic crisis persisted until July 2020 (Centeno and Marquez 2020). More seriously, the research of Bakar and Rosbi (2020) also predicted that global tourism might collapse under the impact of COVID-19 if there were no suitable measures implemented. Recently, there have been a number of publications dealing with global tourism prospects in the post-COVID-19 period. According to Chang et al. (2020), there should be a critical transformation of global sustainable tourism development in order to better recover in post-COVID-19 crisis challenges. Higgins-Desbiolles (2020) proposed the empowerment of local tourism communities, which could be seen as an important transformation moving towards sustainable development.

Cheer (2020) further suggested a concept called "human flourishing", which offers a potential approach to alleviate the pandemic impact on tourism communities and requires current tourism development in a more sustainable transformation. On the 6th of August 2020, the *Journal of Sustainable Tourism* announced an article titled "The war of over tourism" that stressed the challenges to sustainable tourism in the tourism academy after COVID-19 (Higgins-Desbiolles 2020). This report analyzes the challenges faced by academia and the global tourism industry in the post-COVID-19 era. Thus, it can be seen that there is increasingly attention paid among tourism scholars to the impact of the COVID-19 crisis, because this global pandemic may cause long-term consequences, given that the pandemic and responses to it remain ongoing and dynamic. Therefore, it is critical to have a general assessment of the pandemic impacts on different sectors and businesses of a tourism destination, which facilitates a more effective resilience approach.

The impact of COVID-19 on the tourism industry in Vietnam is currently a major concern, because the crisis has entailed a wide range of economic downturn and profound effects on the livelihood of citizens. According to Quang et al. (2020), Vietnam is one of the top 10 fastest growing tourism industries globally, although tourist arrivals sharply dropped by around 22% in early February 2020. More specifically, tourism business revenue fell sharply during just the first 3 months of 2020, representing a VND 143.6 billion loss, because all tourism activities are inter-related (Phạm et al. 2020). In terms of job loss, it is estimated that around 98% of workers in Vietnam's tourism-related businesses left their jobs because of the pandemic (Quang et al. 2020; Tô and Bùi 2020). The authors also indicted that the rate of tourism business suspension or shutdown were increasingly popular among medium and small businesses, especially.

Lê Kim Anh (2020a) and Tô and Bùi (2020) conducted a statistical analysis of the impacts of COVID-19 on Vietnam's tourism industry, including reductions in international

tourist arrival, accommodation establishment closures, increasing unemployment and declining tourism revenue. The authors have proposed key measures targeting expanding marketing, supporting businesses and simplifying immigration procedures. According to analysis from the University of Economics and Finance, the best scenario was that Vietnamese tourism could recover in a 'V-shape', which means a sharp drop and then bouncing back immediately, equivalent to the original decline, or a 'U-shape', depending on the speed of control and economic recovery on the national and global scale. The statistics indicated that the Vietnamese domestic tourism market has recovered first, followed by the Chinese tourist market, Asian tourist arrivals and finally the Western industry. However, many initial articles in newspapers including the Vietnam Communist Party (Trường Đại học Kinh Tế Quốc Dân (Đ.H) 2020), Nhân Dân (Anh 2020b), Vietnam Pictorial (TTXVN/VNP 2020) and by researchers (Giang et al. 2020; Long and Uyên 2020; Thang 2020) have raised attention about the impact of COVID-19 on the tourism business; some recent papers (Ngoc Su et al. 2021; Tri 2021) have pointed out some practical solutions to revitalize the tourism industry in Vietnam. In spite of the increasing concern among tourism scholars in Vietnam, there has been a lack of systematic and in-depth analyses exploring the overall impact of the pandemic on the tourism industry across different tourism sectors of a tourism destination. Thus, this study aims to provide an insight into the total impact of COVID-19 on the tourism industry by employing both secondary data and primary qualitative data collection to reflect the current problems facing tourism stakeholders in Vietnam. These sources of data used as the basis for systematic and in-depth analysis in this study will be further discussed in the next section.

## 3. Research Setting

Can Tho is one of the five cities directly under the Central Government of Vietnam and is the most modern and developed city in the Mekong Delta region of Vietnam. At the same time, it plays a role as a socio-economic, cultural, medical, educational and commercial center of the southwestern region. This city encompasses a large area of 1400 km$^2$, with nine districts and a total population of around 1.3 million people. Can Tho converges many favorable factors for successful development of the tourism industry. Geographically, it is located in a traffic hub linking the provinces in the region. Can Tho has various tourism resource advantages, ranging from natural to man-made tourism assets, which enables this destination to deliver a wide range of tourism experiences to both domestic and international tourists.

Before the pandemic, Can Tho city has welcomed more than 8.8 million visitors, earning more than VND 4435 billion, which contributed nearly 5% of the city's GDP (Ái Lam 2020). The development of tourism in Can Tho city created jobs for 39,300 workers in 2019, of which the numbers of on-duty workers and indirect employees made up 13,100 and 26,200, respectively. To meet the tourism needs prior to COVID-19, Can Tho city had established more than 200 restaurants and catering establishments, 275 tourist accommodation establishments, 22 family-own tourist attractions, 19 home-stays and 59 international and domestic travel businesses. Before COVID-19, the tourism industry of Can Tho city aimed to perform many tasks and solutions to develop the local tourism industry into a key economic sector of the locality. In this regard, the tourism industry attempted to establish an attractive destination image by improving the tourism infrastructure and diversifying the hospitality businesses related to accommodation, dining establishment, sightseeing, tour guiding and so on.

Unfortunately, the ongoing waves of the COVID-19 pandemic have led to collapse of many different sectors of the local tourism system. First and foremost, the manifestation was the decrease in the number of customers for tourism service businesses. According to the statistics, the local tourism enterprises had faced a significant decrease in the number of customers from 30 to 380,000 people, with an average of 11,234.4 people. In terms of relative value, the proportion of customers with the lowest, highest and average decrease was 10%, 100%, and 62.3%, respectively. The decline in the number of customers has led to

a drop in the revenue of tourism service businesses in absolute and relative values. In terms of absolute value, the revenue of businesses has decreased by at least VND 1.3 million and a maximum of VND 102 billion. In terms of relative value, the lowest decline in corporate revenue was 15%, whereas the highest constituted a 100% loss, and the average was 61.3%. The customer reduction also meant a decrease in the capacity of using assets and tourism services, which were reported to fluctuate from by least 5% to 100% (see Table 1). The decline in the number of customers also caused the loss of jobs of current employees in the tourism industry because all local businesses have had to downsize human resources to minimize expenses. As a result, it was estimated that the number of employees in each tourism enterprise who were laid off was at least 1, or could have been up to 70 during the COVID-19 pandemic crisis.

**Table 1.** Impacts of the COVID-19 pandemic on tourism in Can Tho city.

| Indicator/Item | Average Decline (Absolute Number) | Average Decline (Relative Proportion) |
|---|---|---|
| Customers | 11,234.4 people | 62.3% |
| Revenue | VND 2.6 billion | 61.3% |
| Capacity of exploitation/use of property/tourism services | - | 57.4% |
| Reduction in employees | 9.3 | 50.3% |

(Source: Enterprise manager interview data in Can Tho city, 2020).

According to Nguyễn Tuấn (2020), it was estimated that tourists to Can Tho city in 2020 would reach 5,605,865 arrivals, down 36.8% over the same period, reaching 60.9% of the year plan. Accommodation tourists numbered 2,020,145 arrivals, down 32.8% over the same period in 2019, reaching 61.4% of the year plan. In particular, international visitors were estimated to constitute 111,420 arrivals, down 72.7% over the same period in 2019, reaching 25.3% of the year plan. Domestic tourists numbered 1,908,725 arrivals, down 26.5% over the same period in 2019, reaching 67% of the year plan. Outbound tourism packages were provided for 5550 tourists to travel abroad, down 77.6% over the same period in 2019, reaching 20.6% of the year plan. Total revenue from tourism was estimated at more than VND 3169 billion, down 28.6% over the same period in 2019, reaching 62.1% of the year plan. According to Ái Ái Lam (2021), in 2020, about 45.9% of direct workers in Can Tho city were forced to leave their jobs in the tourism sector due to the impact of the COVID-19 pandemic. In spite of the highly expected 2020 tourism targets, the first wave of COVID-19 caused a huge decline in the number of international visitors, while the following waves of COVID-19 placed the local tourism industry under much greater pressure.

## 4. Research Methods

Mixed methods can be used as an effective tool in social science research to increase the validity of research findings and meet a particular purpose of tourism research (McKim 2017). McManamny et al. (2015) also identified a critical reason for the common use of mixed approaches for tourism research to overcome objective deficiencies of employing qualitative or quantitative methodologies alone. As such, a mixed method approach was employed in this study to collect both qualitative and quantitative data to meet the data collection objectives. More specifically, qualitative data allowed researchers to measure the impacts of COVID-19 on the tourism industry of the examined case study, whereas the qualitative data enabled the researchers to gain in-depth information regarding the stakeholder's perceptions and attitudes about the pandemic impacts on their business operations.

The research design included two main phases. The first stage began with developing a framework for qualitative data gathering, whereas the second stage focused on the quantitative data collection. This study adopted the theoretical framework of an impact evaluation model on tourism destination developed by Gertler et al. (2016) and

White (2009). Accordingly, the research team then developed a set of questions for the in-depth interviews, suitable for the research setting in Vietnam. A pilot study was conducted with tourism experts at Can Tho University to standardize the final interview questions. Potential participants for the interview were selected carefully by using a purposive sampling method. More specifically, the study recruited 16 representatives from tourism management organizations (e.g., tourism associations, Department of Culture, Sports and Tourism, centers for tourism development), 16 representatives of tourism-related organizations (e.g., Department of Information and Communications, Department of Health, Department of Planning and Investment, Department of Finance), and 8 representatives from local authorities in the city. Accordingly, the interviews were conducted from October to November 2020. For the second stage of data collection, the questionnaire was framed around the key emerging themes from the interviews which were relevant to the research setting. Then, the questionnaire was critically reviewed by the tourism experts at Can Tho University so that proper modification could be made to complete the official questionnaire. As a part of data collection triangulation, a pilot study with 20 tourists was conducted to test the validity and reliability of the measurement scale and content of the questions. The process of quantitative data collection lasted for 4 months, from December 2020 to March 2021.

Regarding the selection and recruitment of the participants, criteria were developed to choose a relevant list of potential participants who were tourism business managers. The first participant was chosen randomly from the list and the subsequent interviewees were recruited by using a snowball sampling method. In order to collect data regarding the impact of the COVID-19 pandemic on the tourism industry of Can Tho city, 280 tourism-related business managers were recruited. More specifically, the majority of respondents were in charge of managing accommodation (28.6%) and tourist attractions (25%), whereas 17.9% of respondents were from both travel agencies and dining establishments, followed by family-owned businesses and homestays with 10.7%. The sample compared to the population is presented in Table 2.

**Table 2.** Sample compared to population.

| Item | Sample | Population (2018) a | Percentage (%) |
|------|--------|---------------------|----------------|
| Accommodations | 80 | 275 | 29.1 |
| Travel agencies | 50 | 59 | 84.7 |
| F & B establishments | 50 | 200 | 25.0 |
| Family-owned business and homestays | 30 | 41 | 73.2 |
| Tourist attractions | 70 | 70 | 100 |

Source: (a) Data provided by the Department of Culture, Sports and Tourism of Can Tho City.

The questionnaire structure consisted of two parts. The first section gathered the respondents' demographic information, whereas the second part focused on exploring five key aspects: (1) the impact of the COVID-19 pandemic on local tourism business; (2) the measures to alleviate the impact and foster tourism business recovery during and after the pandemic; (3) the supportive policies of the local governments; (4) new business opportunities under the COVID-19 pandemic; and (5) corporate recommendations for the local tourism industry. The questionnaire mainly collected quantitative data by using different measurement techniques including multiple-choice questions, 5-point Likert scales (1, totally disagree; 5, totally agree) and open-ended questions to gather the qualitative data.

Regarding the process of data analysis and interpretation, the qualitative data were coded to identify popular themes and categories that are relevant to the research questions. The emerging themes were grouped using NVivo software. For quantitative data analysis, a total of 280 questionnaires were distributed and collected (response rate at 100%) and all the questionnaires were eligible for further analysis. Using IBM SPSS Statistics 20.0, data from the questionnaire were imported and analyzed to generate descriptive statistics and one-way analysis of variance (one-way ANOVA).

## 5. Research Findings

This section presents the key findings regarding the general impacts of the COVID-19 pandemic on the local tourism industry in Can Tho city, and particularly highlights the vulnerability of different types of local tourism-related businesses, and the collective responses from key stakeholders in coping with the pandemic.

### 5.1. How Serious Is It?

The COVID-19 pandemic has generally damaged the tourism industry in Can Tho city. The sharp decline in the number of customers, especially international visitors, has made many tourist service establishments close or suspend their business operations (Figure 1). Through the interviews with the representatives of the Department of Culture, Sports and Tourism of Can Tho City, about 20% of the tourism service businesses in the city and many tourism- and hospitality-related enterprises have had to close their businesses, which has also meant a huge loss in revenue (business, government, employees), loss of jobs, confusion and burden on society (businesses, employees) and negative impacts on local socio-economic development in both the short and long term (Figure 2).

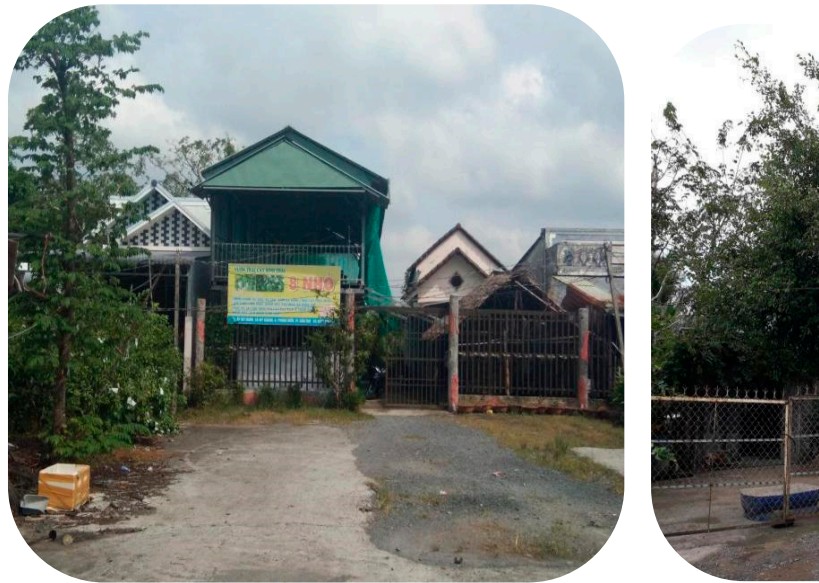 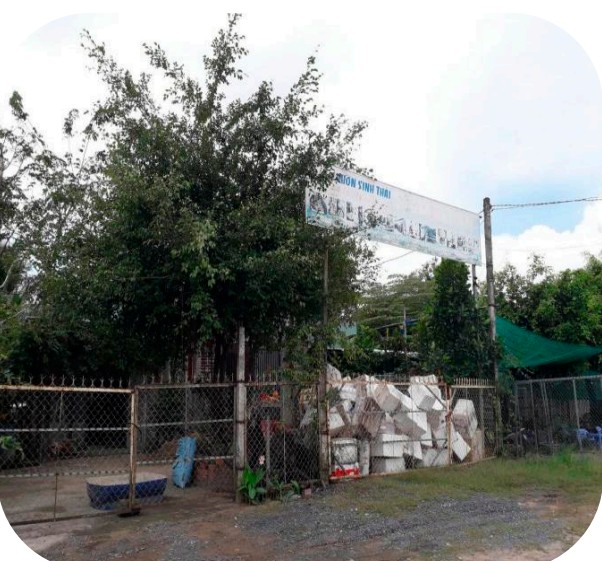

**Figure 1.** Tourist attractions that stopped doing business during the COVID-19 pandemic. (Source: Photos taken by the research team, 2020).

The majority of the managers of tourism service businesses in this study believed that the COVID-19 pandemic has impacted local business either at a serious or very serious level, at 49.3% and 40%, respectively. According to the statistics, very few enterprises admitted that the COVID-19 pandemic has had moderate or non-serious impacts on their business, at 10.4% and 0.4%, respectively. This finding affirmed the serious impact of the pandemic on the tourism and hospitality industry (Panzone et al. 2021).

According to the data provided directly from the Department of Culture, Sports and Tourism of Can Tho City, the revenue of businesses has decreased significantly compared to the same period in 2019. More specifically, the revenue of accommodation establishments decreased on average from 50% to 90%, whereas the revenues of travel agencies and dining establishments decreased by 90% and around 60–90%, respectively.

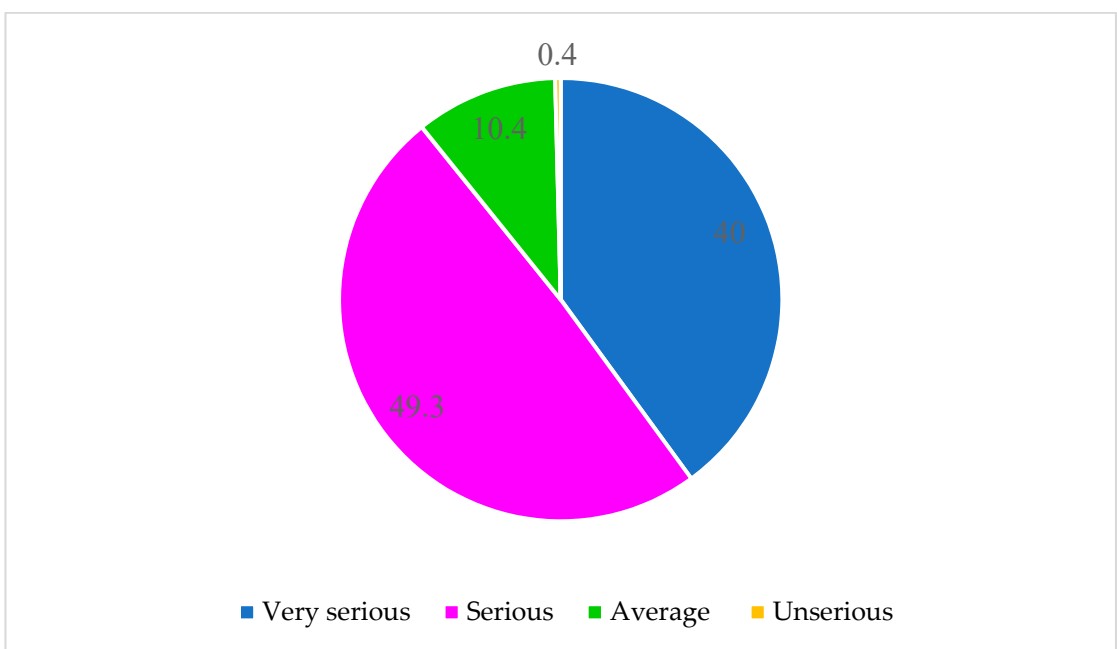

**Figure 2.** The perceived impact of the COVID-19 pandemic on tourism service businesses in Can Tho. (Source: Enterprise manager interview data in Can Tho city, 2020).

The results of qualitative interviews with many stakeholders involved in the tourism business in Can Tho generally reflected their impuissant responses to the COVID-19 pandemic. The respondents stressed that the pandemic has caused a crisis in the tourism industry of Can Tho city, with extremely heavy losses. A representative of Can Tho City Tourism Association said:

*"The COVID-19 pandemic has had a great impact on the tourism industry of Can Tho city, especially the travel agencies, accommodations, and tourist destinations and transport services. Many tourism businesses were on the brink of bankruptcy, causing employees losing their jobs and creating a burden for the society".*

In order to overcome difficulties during and after the pandemic, the findings from the survey indicated that all local business owners were aware of the importance of self-adjustment rather than dependence on the limited financial support from the government. Some common measures to deal with the pandemic consequences associated with renting premises, paying staff salaries and operating costs have been initiated. In this sense, a lot of companies chose to reduce staff numbers or close a part of their business temporarily, whereas the others adjusted their target markets and re-structured their tourism products or services. These are supposed to be temporary solutions for local businesses:

*"We could not just sit there and wait for the financial support from the local and central government. Although we were first passive at the first wave of pandemic, we are now proactive to prepare possible scenarios and have our own measures. Indeed, we are lucky not to close our business but not sure if lockdowns or another wave of COVID-19 occurs".*

*(A representative of a tourism enterprise in Can Tho city)*

According to a representative of the Can Tho City Tourism Development Center, the COVID-19 pandemic has caused a domino effect in the entire tourism economy. As a result, a few surviving tourism enterprises have been unable to operate their business as usual due to a lack of other important suppliers.

*"A half of the businesses in the area were temporarily closed; some businesses operated moderately, over 70% of workers in the tourism industry were affected and unemployed... they had to change their jobs. It can be said that COVID-19 has been stifling the tourism industry, not only in Can Tho city but also nationwide."*

According to the respondents, the domino effect of the pandemic crisis in Can Tho city almost froze its tourism economy, leading to devastating consequences. Evidently, around 63% of businesses in Can Tho city stopped operating; only 16% of businesses could remain open, but had to cut 14% of their operation scale and employees. On the other hand, with the consequences of the decline in customers and revenue, the remaining 7% of businesses had to dissolve when there was not enough funding to maintain operations. A representative of the local government of Cai Rang district projected a dark scenario for domestic tourism in the region:

> "COVID-19 has been and will continue to be a huge challenge for the tourism industry in Can Tho in particular and the entire tourism industry in our country in general". This has caused a great pressure on the social economy when the laborers lost their jobs, the socioeconomic burden increased rapidly, and the job creation problem was under pressure. From that we can see the impact of the pandemic on the tourism industry and the economy. I think that the epidemic crisis is extremely serious."

*5.2. COVID-19 Impact and the Vulnerability of Tourism Business Sectors*

The research findings generally indicated differences in the impact of the COVID-19 pandemic on some tourism sectors and the types of tourism-related enterprises. Each type of tourism business with different business characteristics has been affected differently by the pandemic. According to the framework of this study, one of the key research objectives was to explore if there was any difference in the impact of the pandemic on business operations of the restaurants, accommodations, travel agencies, tourist destinations and family-owned tourist attractions. The results of data analysis by one-factor variance analysis showed that there was a statistically significant difference in the impact of the COVID-19 pandemic on some key business activities in terms of the percentage of customer decline, total decline, the proportion of revenue and the proportion of employee reduction (See Table 3).

**Table 3.** ANOVA results.

| Item | | df | F | *p* |
|---|---|---|---|---|
| | Between Groups | 4 | 0.992 | 0.415 |
| Customer reduction | Within Groups | 102 | | |
| | Total | 106 | | |
| | Between Groups | 4 | 7.339 | 0.000 |
| Percentage of customer reduction | Within Groups | 274 | | |
| | Total | 278 | | |
| | Between Groups | 4 | 2.466 | 0.052 |
| Revenue reduction | Within Groups | 79 | | |
| | Total | 83 | | |
| | Between Groups | 4 | 4.041 | 0.003 |
| Percentage of revenue reduction | Within Groups | 257 | | |
| | Total | 261 | | |
| Percentage of reduction in capacity of | Between Groups | 4 | 1.665 | 0.158 |
| exploitation, property use and tourism | Within Groups | 263 | | |
| services | Total | 267 | | |
| | Between Groups | 4 | 1.737 | 0.146 |
| Number of employees to be laid off | Within Groups | 127 | | |
| | Total | 131 | | |
| | Between Groups | 4 | 11.133 | 0.000 |
| Percentage of employees to be laid off | Within Groups | 127 | | |
| | Total | 131 | | |

(Source: Enterprise manager interview data in Can Tho city, 2020).

According to Table 4, the COVID-19 pandemic has caused a decrease in the percentage of customers and the percentage of employees being laid off, differing from business to business. In terms of the proportion of customer decline, tourist fruit orchard houses were

the most affected (68.5%), followed by other tourist attractions (67.8%). The other types of tourism businesses with smaller decreases in the number of customers than travel agencies made up 65.8%, compared to accommodation establishments and dining establishments at 58.4% and 53.4%, respectively. Regarding the proportion of employees being laid off, large tourism sites and family owned-tourist attractions were the two organizations with the greatest proportion of employees being laid off at 74.3% and 63%, respectively. The other tourism-related businesses including accommodation, tour operators and dinning establishment experienced similar impacts, at 46.4%, 39.5% and 39.2%, respectively.

**Table 4.** Differences of COVID-19 impacts on business performance of firms in Can Tho city.

| Item | | N | Mean |
|---|---|---|---|
| Percentage of customer reduction | Accommodation | 80 | 58.4 |
| | Travel agency | 49 | 65.8 |
| | Dining establishment | 50 | 53.4 |
| | Tourist attraction | 70 | 67.8 |
| | Tourism fruit orchard | 30 | 68.5 |
| Revenue reduction (VND millions) | Accommodation | 20 | 2337.7 |
| | Travel agency | 13 | 11,336.9 |
| | Dining establishment | 9 | 1577.9 |
| | Tourist attraction | 30 | 263.2 |
| | Tourism fruit orchard | 12 | 240.5 |
| Percentage of revenue reduction | Accommodation | 80 | 58.3 |
| | Travel agency | 48 | 64.2 |
| | Dining establishment | 48 | 54.7 |
| | Tourist attraction | 63 | 66.5 |
| | Tourism fruit orchard | 23 | 65.3 |
| Percentage of employees to be laid off | Accommodation | 34 | 46.4 |
| | Travel agency | 20 | 39.5 |
| | Dining establishment | 40 | 39.2 |
| | Tourist attraction | 28 | 74.3 |
| | Tourism fruit orchard | 10 | 63.0 |

(Source: Data collected from interviewing tourism enterprises in Can Tho city, Vietnam, 2020).

The above findings generally indicate the decline in revenue for different types of businesses. Unexpectedly, tourist destinations and family-owned tourist attractions experienced the greatest decline in revenue at 66.5% and 65.3%, respectively. Meanwhile, 58.3% of accommodation establishments and 54.7% of food establishments witnessed a decrease in revenue. At 90% confidence, the revenue of different types of tourism business experienced different drops. The travel agencies, accommodation establishments, and food and beverage establishments were those with the greatest decrease in total revenue, at VND 11,336.9 million, VND 2337.9 million and VND 1577.9 million, respectively. Meanwhile, the tourist destinations and family-owned attractions saw relative declines in total revenue of VND 263.2 million and VND 240.5 million, respectively.

## 6. Discussion

This section addresses the research question by revisiting the significant findings and providing in-depth discussion regarding the overall impact of COVID-19 on the tourism industry, particularly highlighting how vulnerable local tourism businesses suffered from multiple waves of the pandemic.

### 6.1. Overall Impact of the Pandemic on the Local Tourism Industry

The COVID-19 pandemic generally caused a standstill of tourism development in developing cities in Vietnam during the first wave. The findings in this study indicated that this paralysis has resulted in a plethora of negative impacts on different sectors of the local tourism industry. The first wave of the pandemic placed local businesses under such intensive pressure. Nearly 90% of tourism-related businesses faced economic impacts at

serious or very serious levels, with 20% of tourism-related enterprises having to temporarily close their business. This implies that a tourism-dependent city might be under much more pressure because of the huge tourism revenue losses. The evidence from this research revealed increasing concerns of the temporary closure or shutdown of many different genres of tourism enterprises and suppliers due to the sharp drop of tourism demand, which has directly caused supply chain disruption of the whole tourism system. This also means that even surviving enterprises during the waves of the pandemic might find it difficult to recover their financial capabilities due to the limitations of local tourism business activities.

Evidently, local enterprises were found to face the increasing revenue losses, fluctuating around 50% to 90% depending on the types of tourism-related businesses. Of which, tourism enterprises were found to suffer the hardest hits, with nearly a 90% economic downturn during the initial waves of the pandemic. However, it was estimated that the real impact of the COVID-19 pandemic on the local tourism industry in this examined case study was even worse (Mekong Delta Tourism Association 2020). Similar research findings by Kuqi et al. (2021) also reconfirmed the tourism business crisis in Kosovo and the huge decline in tourism revenue compared to previous years. Other studies also reported a significant drop in tourist arrivals during the pandemic and their intentions not to travel post-pandemic (Terziyska and Dogramadjieva 2021). This reveals the serious effects of the pandemic on tourism demand at the global scale (Deyshappriya et al. 2021). In general, the pandemic has caused a plethora of significant impacts on different aspects of the local tourism economy and may contribute to a long-lasting crisis for the national tourism system which will be further discussed in the next section.

### 6.2. Domino Effects of COVID-19 on the Local Tourism Industry

The impact of the COVID-19 crisis has leaded to a domino effect in different areas of the local tourism system. The findings in this research reflected the reality of the hardest hits to the local tourism enterprises, many of which were found to be on the edge of bankruptcy. This finding could be similar to that of the research by Wieprow and Gawlik (2021), which also confirmed that the pandemic crisis might entail the collapse of the tourism enterprises in Poland. The same is true for business bankruptcy on a large scale in Argentina due to lingering pandemic impacts (Korstanje 2021). This common phenomenon could be interpreted in many different ways. First of all, the long-lasting lockdowns during multiple waves of COVID-19 have affected all local businesses, especially small enterprises with limited financial capacity. As a result, the small business enterprises in this study were likely to face a possibility of real shutdown, and very few small-sized business enterprises were found to overcome the second wave of the pandemic. A study by Kalogiannidis (2020) also confirmed that small businesses might be at risk and face the practical possibility of bankruptcy due to long lockdowns. Moreover, even small businesses in European countries which benefited from their country's good financial schemes also faced financial crisis, and very few of them could survive repetitive and long-lasting lockdowns (García et al. 2020; Parikh 2020).

In addition, the domino effect also brought indirect consequences to other tourism-related business sectors when tourism demand experienced a free fall as a result of repetitive waves of the pandemic. The tourism managers in this study showed little optimism about post-crisis recovery because of supply chain disruption and the existing barriers. First of all, financial difficulty, which was found to be the most common concern among the tourism enterprises, necessitated better financial support from both local and central government. Evidently, Vietnam's tourism stimulus packages focused more on easing pandemic control restrictions and fostering the domestic demand marketing, whereas the financial measures were inefficiently addressed (Tri 2021). As such, the majority of the surviving tourism enterprises after multiple waves of the pandemic were likely unable to maintain temporary operation in the long run, whereas the medium- and small-sized enterprises had to face more challenges in coping with fiscal measure shortages during the ongoing pandemic. Bartik et al. (2020) suggested that small enterprises had to take their

own measures to overcome the challenges, although the researchers accepted the reality that financial policy responses from local levels might be limited to strengthen the survival ability of local businesses.

The other typical barrier to local business recovery is the governmental policy of quarantine. The tourism managers in this study claimed that any positive case of COVID-19 found at a local business might lead to a temporary shutdown of their business operation, and this influenced their recovery opportunities during the pandemic waves. According to Masondo (2021), the mass collapse of local businesses during the pandemic could be alleviated if the tourism businesses had effective preparation to be more resilient to the impacts and more flexible to adapt to the governmental policy of pandemic prevention and control. Similarly, the Vietnamese policy of pandemic prevention and control was found to have impacted the poor resilience of the local tourism industry in Can Tho, which will be discussed in-depth in the next section.

### 6.3. Poor Resilience to Pandemic Impacts

According to the majority of the research respondents, the governmental policy response to the pandemic might have strongly influenced local business operation and resilience. A lot of tourism enterprise managers explained that the slow and passive response to the pandemic from local and central governments might have contributed to the ineffective responses of the local businesses. This result reconfirmed the previous study by Van Van Nguyen et al. (2020), that Vietnam's central government had a slow response to the first wave of the COVID-19 pandemic, causing difficulties for the tourism industry to cope with the impacts of the pandemic. Moreover, most of tourism enterprises shared their increasing concern about the lack of more effective visions and measures from the government which could orient the local businesses, especially the small- and medium-sized enterprises, to better respond to the pandemic in the long term. According to the statistics, over 63% of the local businesses temporarily suspended operations, most of which were small- and medium-sized travel businesses. This finding reconfirmed the previous studies that the COVID-19 pandemic might have long-term impacts on all tourism-related businesses, and the recovery ability of medium- and small-sized companies is at serious risk (Wieprow and Gawlik 2021). Therefore, the governments at all levels should be more concerned about their leading role toward the goal of pandemic impact alleviation on the tourism industry and initiate prioritized tourism polices which could strengthen the recovery of all tourism-related businesses before it is too late.

Regarding the measures to reduce the impacts of the pandemic on the local tourism businesses, a collaborative approach was believed to better support the tourism enterprises in overcoming hits from the pandemic. The research findings in this study indicated that there have been different levels of suffering of different businesses and tourism enterprises against the pandemic crisis. Therefore, financial support from the government might consider a case-by-case basis (Ministry of Industry and Trade of Vietnam 2021). In addition, each enterprise should initiate their self-adjustment plan in a proactive way, enabling them to overcome waves of COVID-19 impacts. According to Rosli and Jamil (2020), the enterprises with efficient measures against pandemic might have a better chance to cope with difficulties during the crisis. This reconfirmed the significant role of tourism enterprises to revitalize their business capability rather than the dependence on the governmental support.

However, these findings do not mean that governments can underestimate their role towards addressing the most challenging difficulties related to the financial incapability of small enterprises, as discussed above. Therefore, the authority's timely policy responses could be a key factor that strengthen local business resilience. A good example of effective policy responses during the second wave of COVID-19 in Vietnam was that the governmental tourism stimulus package fostered the domestic tourism and saved local enterprises in Can Tho, recovering their destination capacity to around 60%. To some extent, this important evidence elicits a positive belief that the pandemic impacts on tourism businesses

could be alleviated, and tourism-related enterprises could be saved if proper measures are taken (Atalan 2020).

*6.4. Does the Size of Tourism Businesses Matter?*

In response to the impact of the COVID-19 pandemic, tourism enterprises in Can Tho were likely to be the most vulnerable businesses. However, there were some key differences in the vulnerability related to the size of the local enterprises. There are many potential causes to explain this phenomenon. First of all, the tourism companies whose direct revenue mainly relies on tourists might have suffered more due to the matter of fact that tourist arrivals dropped significantly during the pandemic. Moreover, the findings in this study further explained that dependence on a specific tourism offering or target market might have led to financial crisis during lingering lockdowns in Vietnam. As a result, those big companies were likely to lose far more revenue than smaller enterprises.

However, the big company managers in this research supposed that the restructuring of their business model by diversifying tourism products and positioning the target market have enabled them to reduce the economic loss. A potential measure regarding the reduction in workforce was also found to be a temporary solution for all kinds of local businesses. This appears to be understandable, because downsizing the number of employees at tourist attractions has been a common measure for enterprises to alleviate finance pressure to survive during the pandemic (Hamilton 2020). Although big tourism enterprises could restructure their business in the short term, they were found to be more vulnerable in the long term. In general, the above-mentioned findings imply that the COVID-19 pandemic has severely affected the entire tourism industry of Can Tho city and the lingering crisis may lead to an entire collapse of the hospitality and tourism enterprises no matter what their size of business is. In order to prevent worse scenarios, governments at all levels may have offered special support to save different types of tourism enterprises which are almost on the brink of bankruptcy due to domino effects of the pandemic.

## 7. Conclusions

This research has highlighted the overall impacts of the COVID-19 pandemic on different sectors of the tourism industry in a developing city in Vietnam. The study found that there were differences in terms of the levels of impact on local tourism destinations throughout the various COVID-19 waves. The impact of the pandemic on the examined case study has generally caused a wide range of negative impacts in local destinations, including enormous losses in the tourism revenue, significant decreases in the number of both domestic and international tourist arrivals, the inadequacy of destination exploitation capacity, the temporary or permanent closure of tourism-related services, and the rising unemployment rate. In terms of the level of influence, all tourism businesses had suffered from the pandemic, ranging from serious to very serious levels.

However, there has been a difference in the degree of business damage, which can be seen from the percentage of customer decline, the decrease in absolute value, the relative turnover and the proportion of employee redundancies. The research findings also indicated that tourist attractions and tourism enterprises were the key stakeholders facing the biggest decline in the number of customers, the percentage of staff downsizing and the highest proportion of revenue decline. Additionally, tour operators, accommodations and dining establishments experienced the greatest losses in the absolute value of tourism revenue. However, it is noticeable that there was no difference in terms of the damage in each tourism service company in terms of decreases in the number of passengers, the percentage of tourism asset, service capacity, utilization and the absolute value of employee downsizing. Among the businesses, tourism enterprises were found to be the most vulnerable to the pandemic. More seriously, the pandemic crisis has caused a domino effect on different sectors of the local tourism system. Therefore, more than 20% of tourism-related enterprises have had to close their business, whereas the remainder have faced post-pandemic recovery challenges.

In response to the pandemic, the proactive measures initiated by the tourism enterprises were found to be effective, whereas financial solutions from the government were supposed to be critical but inefficient for local tourism revitalization. The effective measures reported by the local businesses included promoting the exploitation of the domestic tourist market, developing new products, ensuring safety for employees and tourists, developing quality tourism human resources, service price reduction, organizing a number of tourism stimulating events with a focus of motivating tourists, implementing capital support policies for each sector of the businesses, prioritizing domestic tourism recovery, ensuring hygiene at tourist sites and destinations and promoting the application of information technology in promoting destination images. In general, the staged model applied for crisis management during the second and the third waves of COVID-19 in Vietnam has been successful. These findings indicated that effective collaboration among all key stakeholders might contribute to alleviating the impacts of the pandemic.

Theoretically, this study provides a potential enrichment to the literature related to the global pandemic impacts and crisis management. This study indicates that there will be no "one-size-fits-all" approach to global pandemic crisis management such as for COVID-19. Therefore, this research adds a novel theoretical contribution to crisis- and disaster-management models for tourism destinations which should be flexibly staged and multiple strategic action plans integrated, because this global pandemic will cause lingering and repetitive impacts on tourism destinations. Moreover, the study sheds light on the roles of key stakeholders at different stages of pandemic, which contributes to building a strategic and holistic framework in effectively coping with pandemic impacts on different stages of the pandemic. Finally, the study offers better understanding of the vulnerability of tourism business and collective measures for instant pandemic responses in a more sustainable way.

Importantly, this study provides several practical implications for the tourism industry under the pressure of COVID-19 crisis. First of all, it provides destination managers better understanding about the strategic management to deal with the pandemic impacts and adjust potential measures to reduce the crisis consequences. In this regard, due to the long-lasting pandemic impacts, the surviving tourism enterprises should be more aware of the difficulties ahead. As such, the tourism enterprises should be adaptive to the impacts of the pandemic by restructuring their tourism products and target markets in order to survive during multiple waves of COVID-19. Importantly, tourism businesses should be more proactive by initiating their own measures to achieve post-crisis revitalization instead of passively relying on governmental support. The findings in this study also imply that the tourism system should have various action plans prepared to deal with similar global disasters in the future. It is also suggested that governments may need to take the leading role in emergent crisis management and address the conundrums associated with financial crises which directly affect destination resilience and tourism enterprises' post-pandemic recovery prospects.

Although this study has attempted to provide both theoretical and practical implications regarding the impact of the COVID-19 pandemic on the tourism industry in a developing country context, there are still some limitations available for future research. The sample size of this study may not be representative for the whole tourism industry in Vietnam, but provides interesting ideas for similar research in different settings where the tourism industries may have also suffered multiple waves of COVID-19. In addition, potential research may investigate the effectiveness of governmental policy responses to the survival of the tourism industry and revitalization during and after pandemic. Last but not least, prospective studies may consider exploring key factors and stakeholders affecting the success of domestic tourism resilience during the pandemic.

**Author Contributions:** Data curation, G.V.H.D.; Formal analysis, D.V.H., N.T.N. and G.V.H.D.; Funding acquisition, D.V.H.; Investigation, T.T.K.T., N.T.N. and C.N.D.; Methodology, L.H.D.; Project administration, D.V.H.; Resources, C.N.D.; Supervision, L.H.D.; Validation, T.T.K.T.; Writing—original draft, D.V.H.; Writing—review & editing, L.H.D. All authors have read and agreed to the published version of the manuscript.

**Funding:** This research was funded by the Vietnam National Foundation for Science and Technology Development (NAFOSTED) under grant number: ĐXTN-2020.07.

**Conflicts of Interest:** The authors declare no conflict of interest.

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
