# Peer review of "The COVID-19 Pandemic and Its Impacts on Tourism Business in a Developing City: Insight from Vietnam"

_economies, doi:10.3390/economies9040172_

Round 1
Reviewer 1 Report
The paper shows important issue for tourism sector nowadays – pandemic COVID-19 and its impacts on tourism business. The results of this research can be interesting to tourism industry. That is why it should be interested to the readers, especially for the entrepreneurship from tourism sector. In my opinion there is a significance of providing such kind of research nowadays. The background includes actual references. The research design is appropriate. The methods are adequately described. In the last section I suggest to divide it into 2 parts: Discussion and Conclusions. In the discussion section, In the Discussion section state other similar findings from the literature and compare with the ones you have. Conclusion are very poor written, the results should be recalled and other similar ones and specify the theoretical and practical implications this study has, also the novelty that adds on the field.
Author Response
Response to Reviewer 1 Comments
First, I would like to take this opportunity to express my sincere appreciation for your time and critical comments on my paper. I have carefully clarified all the comments as you can see them below.
Point 1: In the last section I suggest to divide it into 2 parts: Discussion and Conclusions.
Response 1: Yes, I have split the section into Discussion and Conclusions relocated in the revised manuscript at page 11-13 and 13-15 respectively.
Point 2: In the discussion section, In the Discussion section state other similar findings from the literature and compare with the ones you have.
Response 2: Yes, I have revisited the research findings compared with previous studies and further discuss the key findings to address the research question. Please see page 11-13.
Point 3: Conclusion are very poor written, the results should be recalled and other similar ones and specify the theoretical and practical implications this study has, also the novelty that adds on the field
Response 3: Yes, I have modified and improved the conclusion section. Regarding the theoretical implications, the novel contribution to crisis-and-disaster management model for tourism destinations have been highlighted (please see the Conclusion section- paragraph 3). I have justified why impact management model should be flexibly staged, and integrate multiple strategic action plans since such global pandemic may have lingering and repetitive impacts on tourism destinations.
Concerning the practical contribution, I have stressed how this study benefit the tourism destination managers, tourism enterprises and other related business owners to respond to the impact of the pandemic in an effective way (please see the Conclusion section- paragraph 4)
Reviewer 2 Report
Dear author/s,
Thank you for the opportunity of reading the paper “Pandemic and its impacts on tourism business in a developing city: An insight from Vietnam”. The topic is interesting and the research offers valuable information about impact of Covid-19 pandemic on tourism sector. However in order to increase the quality of the paper there are a few recommendations:
- This paper has no research questions of research hypotheses.
- I suggest presenting the research area before the methodology.
- Please clarify if the questions from the second part were open questions.
- I recommend separating the results part of the discussions part.
- Which are the managerial implications of your study?
Others:
Pay attention to decimal separator.
Good luck!
Author Response
Response to Reviewer 2 Comments
First, I would like to take this opportunity to express my sincere appreciation for your time and critical comments on my paper. I have carefully clarified all the comments as you can see them below.
Point 1: This paper has no research questions of research hypotheses
Response 1: We aim to collect main qualitative data to explore the research phenomenon so the quantitative method is just complementary to our research. I totally agree to add the research questions (please see the Introduction-page 2)
Point 2: I suggest presenting the research area before the methodology.
Response 2: Yes, I have relocated the Research setting before the Methodology. Please see page 4-5).
Point 3: Please clarify if the questions from the second part were open questions.
Response 3: Yes, types of questions and information collected have been clarified and added (please see it at early page 6)
Point 4: I recommend separating the results part of the discussions part.
Response 4: Yes, I totally agree. The Results and Discussion have been divided into 2 separate sections (Please see Results and Discussion section at page 7 and 11 respectively)
Point 5: Which are the managerial implications of your study?
Response 5: I have clarified the theoretical and practical implications of my study (Please see Conclusions section, page 13, paragraph 3-4)
Point 6: Pay attention to decimal separator.
Response 6: Yes, I have double checked the decimal separator
Reviewer 3 Report
The article presented for review is very interesting and presents an important aspect of research related to the impact of the pandemic on the tourism market.
Mixed methods were chosen to achieve the goal set by the authors, which gave an interesting final result.
There are many studies on the impact of a pandemic on specific local markets, which may allow for the development of general rules of conduct in the event of such a threat on the entire tourism sector. It is very important to analyze the impact of this situation on tourism enterprises and, more broadly, on other entities cooperating with them. One cannot forget about the domino effect and the negative impact of reducing tourism demand not only on the industry but also on other non-touristic entities.
It is worth doing further research to analyze what the future of the local tourism industry and these non-touristic entities will be after the pandemic.
Author Response
Dear Madam/ Sir,
We would like to take this opportunity to express my sincere appreciation for your time and critical comments on my paper.
Thank you so much again
Kind regards
Research team
Round 2
Reviewer 2 Report
Dear author/s,
thank you for the improved version of the manuscript. The paper is now suitable for being published.
Good luck!